# Civilized Muscles: Building a Powerful Body as a Vehicle for Social Status and Identity Formation

Ask Vest Christiansen 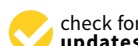

Department of Public Health, Aarhus University, 8000 Aarhus, Denmark; avc@ph.au.dk

**Abstract:** This paper explored the relationship between having a muscular body and identity formation in young men. Theoretically, it was built on evolutionary psychology; empirically, it drew on the author's research into young men's use of anabolic-androgenic steroids in gym settings. The questions I addressed were the following: First, why does the building of a muscular body through weight and strength training appeal to young men who have not yet found their place in the societal hierarchy? Second, what identity-related consequences does it have for them, when the size and posture of their body changes? First, the paper outlined some important aspects of the civilizing process and evolutionary psychology in order to offer an explanation on how and why brute force has been marginalized in today's society, while the strong body continues to appeal to us. Then followed an explanation of the concept of identity used in this context. Hereafter, it was examined how building a more muscular body influences the young men and their relationship with their surroundings. Next, an underlying alternative understanding of health that may influence young men's decision to use anabolic steroids was discussed. The article concluded with some remarks on the body's impact on identity in a time where a strong build no longer has any practical importance in our lives.

**Keywords:** gym culture; identity formation; anabolic androgenic steroids; social status; evolutionary psychology

## 1. Introduction

He, who can beat the others, will win respect. He qualifies for a position in the upper echelons of the social hierarchy. If you are a wimp, on the other hand, you have to accept a lower place in the pecking order. The strong ones go to choose first, while the weak must take a subservient position in the back of the line.

The viewpoint that using violence, or merely threatening using violence, should dictate our social position, fundamentally opposes most peoples' views (Pinker 2011). In the history of mankind, it is, however, a relatively new phenomenon that a man's capacity for violence is unrelated to his position in the social hierarchy. Yet, while the social position in the early 21st century is not expressed through muscles and strength, but rather by means of education, jobs, houses, clothing, cars, and travels, such symbols are not easily accessible for young men. Thus, for many of them, the male body and the signals it transmits to the surroundings about its strength continue to play a crucial role in how they feel positioned socially.

In this paper, I explored why and how strength and muscles continue to be a lure for young men.[1] The questions I addressed were the following: First, why does the building of a muscular body through weight and strength training appeal to young men who have not yet found their place in the societal hierarchy? Second, what identity-related consequences does it have for them, when the size and posture of their body changes? To answer this, I began by outlining some important aspects of first the civilizing process and second evolutionary psychology in order to offer an explanation of how and why brute force has been marginalized in today's society, while the strong body continues to appeal to us. I then explained what I mean by identity in this context. Hereafter, by means of examples from my research among gym-goers, I showed how bodily transformations might influence men's self-image and their interaction with the surroundings. Next, I discussed an underlying alternative understanding of health that might affect young men's decision to use anabolic-androgenic steroids (AAS). I concluded with some remarks on the body's impact on identity in a time where a strong build no longer has any practical importance in our lives[2]. First, however, a few words on method and theory.

## 2. Method and Theoretical Background

This article sprang from a larger research project on gym culture and identity and performance-enhancing drugs among men aged 21–41 (Christiansen forthcoming). Part of this research involved long, qualitative interviews with men who have used or strongly considered using AAS[3] and other image and performance-enhancing drugs (IPEDs). While I drew on these interviews, the article is not to be thought of like an interview study per se. Likewise, the quotes I presented from the interviews I have conducted are not to be considered an exhaustive account of my empirical material or on the specific topic as such but are meant to support the more fundamental ideas I discussed. Further, while I am familiar with the extensive literature and qualitative research in the area of IPED use and gym culture, it was not the purpose here to address that body of research directly. Rather, based on my readings of this research, the aim was to explore some fundamental (albeit not always conscious) motivations connected to young men's strive for muscularity that has not been given much attention in the academic literature.

As mentioned, the article relates to my research in gym environments. The overall research question I addressed for this was: "What it is that motivates young men to use AAS and other IPEDs, and what do the potentially dramatic changes in muscularity mean for their self-image and identity formation? The research included 14 long, qualitative semi-structured interviews with men aged 21–41 years. Interviewees were recruited partly via posts with an invitation and a link with further information on two websites popular among fitness enthusiasts, and partly via leaflets on the counter of gyms known to have serious weight-lifting members. Finally, some informants were recruited by a snowball-method where one informant suggested another. All interviewees were white, Caucasian, and identified as heterosexual. The interviews were conducted in accordance with the guidelines for semi-structured qualitative interviews (Creswell 2013; Kvale and Brinkmann 2009) and lasted between 1 h 50 m and 2 h 30 m. They were subsequently transcribed in full and coded for content, using an inductive coding approach and qualitative software (Atlas.ti).

---

[1]　This articles is an edited and translated version of a Danish language article appearing in an anthology edited by Rasmus Bysted Møller and Jan Kahr Sørensen (Christiansen 2019). It thus contains sections and quotes from informants that also appears in the article and in my forthcoming book (Christiansen forthcoming).

[2]　This article focusses on men. It does not discuss women's relationship with their bodies and identity formation. Neither the empirical material nor the literature I have used allows for such exploration.

[3]　Anabolic-androgenic steroids are a group of synthetically manufactured hormones that simulate the effects of the male sex hormone, testosterone. These substances both have tissue building (anabolic) and masculinizing (androgenic) effects but are mainly used because of their ability to increase muscle growth and strength. In addition to their performance-enhancing capacities, they can also result in a number of side effects, such as increased blood pressure, acne, testicular atrophy, and mood changes (Pope et al. 2014).

In this paper, I have only used quotes from interviews with five informants: Anton, Martin, Jesper, Kenneth, and Kasper who were between 22 and 27 years of age at the time of the interview. All of them lifted weights with the primary motivation to get stronger, more muscled, and to attain a more attractive body. None trained for a specific sport outside the gym. Anton, Martin, and Jesper had used AAS to become bigger and stronger, Kenneth and Kasper had strongly considered using, but decided not to.

As noted, this paper examined how the male body and the signals it transmits to the surroundings of its strength play a crucial role in how young men feel positioned socially. A man's strength may appear irrelevant for his position in the social hierarchy in today's western, civilized societies, but—and this is the hypothesis my argument rests on—evolution has equipped us with a psyche that is receptive to the signals the strong body transmits to its surroundings (Kenrick et al. 2004; Petersen et al. 2013; Etcoff 1999). This is why we treat stronger men with greater respect than weaker men, and why there is still an attraction in becoming strong and muscular, if you are, or feel you are, weak. This is the case, even if the acquired strength is not used for anything. In examining this assumption, I built on evolutionary psychology. In the context of evolutionary psychology, gender-based differences in norms and behaviors reflect adaptations to problems our distant male and female ancestors regularly had to face. The basic premise for studying gender differences in light of humankind's gradual development is that our brains have evolved in certain surroundings to cope with certain conditions that differ immensely from human surroundings and conditions today (Kenrick et al. 2004). This does not mean that culture is irrelevant but that biology is a precondition for human culture. As the American psychologist Douglas Kenrick and his colleagues argue; in human biology, we can trace back a number of universal, fundamental features concerning behaviors and norms, but, of course, these are expressed in ways that are culturally specific (Kenrick et al. 2004).

## 3. Civilized Muscles

As demonstrated by the German sociologist Norbert Elias (1897–1990), there is a strong link between the civilizing process on the one hand side and our ability to exert self-control and delay of gratification on the other. It goes for our ability to restrain ourselves so you "don't relieve yourself in front of ladies, or before doors or windows of court chambers"[4], or our capacity to control impulses, suppress aggressions, and not start a fight with (or cut the nose off) an offender, demonstrating that we have become better at this as the civilization process has progressed. Or rather, civilization progressed as we became better at self-control, inhibiting our impulses and began anticipating the long-term consequences of our actions[5]. More civilization means less violence and vice versa. Comparisons across different cultures show that the lesser state-formation a society or culture has, the greater the likelihood that men in that culture will suffer a violent death (Pinker 2011). Thus, the idea of the noble savage, known from, among others, the philosopher Jean–Jacques Rousseau (1712–1778) and the anthropologist Margaret Mead (1901–1978), who lives in simple societies in peaceful harmony, is more a romanticized idea than an accurate anthropological description. A critical element in the civilizing process and a central reason why violence largely has disappeared from civilized societies is that it has

---

[4] As the advisory is presented in one of the etiquette books from the Middle Ages that Elias references: Desiderius Erasmus' *On Civility in Boys*. Here quoted from Pinker (2011, p. 83).

[5] Naturally, the process is far more complex than space allows me to explain here. The elements are tied together and mutually dependent. Elias proposed two triggers to get the process started. The first was the consolidation of efficient states with centralized monarchies. Hereby knights, warlords, chiefs, barons, noblemen, and people, in general, passed over the power to a third party, so retaliation and revenge were no longer necessary when, and if, one was attacked or offended. Instead of taking matters into one's own hands and retaliate, it was the state's duty to punish the aggressor. The second trigger was the economic revolution that arose with the division of labor, the formation of the market, and the improved infrastructure. Hereby, the zero-sum game of the Middle Ages was replaced by a positive-sum game where both parties gained from the transaction. You do not fight the ones you trade with; rather, you make an effort to behave and be of good appearance, so you can enhance your sales opportunities and increase your turnover. If one can trust that the state will punish violence, it is a better strategy to trade with the neighboring village rather than plundering it (Pinker 2011, pp. 88–94).

been monopolized. For example, in Scandinavia—one of the least violent regions of the world—one may be led to believe that the primary task of the police is to regulate traffic, so a delivery truck does not hit a commuting cyclist. However, it is indeed due to the police's monopoly of violence that the weak do not have to fear the strong.

Conversely, in earlier times, when violence was present in many aspects of existence, the weak had every reason to hold such fear. While the group's power was essential in rivalry with external enemies, a man's strength was decisive for his position in the internal hierarchy (Diamond 1997; Pinker 2011). Yet, one's strength had to be used wisely. From an evolutionary perspective, it is too costly to turn aggression into violence, if the likelihood of coming unscathed through a confrontation is small. Therefore, those who did well in former societies, who survived and had children who themselves survived, were those who, based on visual cues, were able to read the hierarchies, so they only ended up in violent confrontation when it was necessary or paid off. Having such a capability means that competition can be solved through behavior based on observation and social learning on hierarchies, rather than through constant fighting. This reduces the risk for all members of the group and makes repetitive fighting unnecessary (Herbert 2015). However, it also has the consequence that one has to accept that resources are unevenly distributed. Yet, this does not mean that social hierarchies are static. As Joe Herbert, professor emeritus at Cambridge University, explains, studies of both humans and primates show that a male's position is constantly monitored and tested by other males. Social hierarchies reduce the amount of aggression and violence, but the tension between the males remains a constant factor (Herbert 2015, p. 80).

As noted, violence in the Middle Ages was much more prevalent than it is today. Still, violence in Europe in the Middle Ages was much lower than it was and is in non-state societies. In his book *The Better Angels of Our Nature—A History of Violence and Humanity,* Harvard psychologist Steven Pinker demonstrates how the human species' retreat from violence evolved through six large historical trends (Pinker 2011). Especially, the increased state formation emerging during what historians sometimes call the Humanitarian Revolution in the 17th and 18th centuries made the prospect of violent death manifold smaller. The trend continued after World War II and has been characterized by a growing revulsion against aggression and the use of violence—even on a small scale (Pinker 2011).

This augmented sensitivity is evident when confronted with advertisements of strength training that are only decades old. In the 1940s and a few decades onward, strength training equipment could be marketed with a narrative explaining that among the advantages of being strong was the ability to whack someone who was being a bully. Famous are the advertisements for Charles Atlas and his self-improvement program that held a particular appeal for insecure teenage boys. One of these ads, which appeared in several versions in magazines and comic books from the 1940s to 1960s, features a cartoon strip of a young man being harassed on the beach by a sand-kicking bully while his girlfriend looks on passively. Back in his room, he angrily kicks a chair, "gambles a stamp", and orders physical-fitness material from Charles Atlas. "Later", he returns to the beach with his new, impressive physique. It's pay-back time: He punches the bully in full public view, and his girlfriend and the other beach-goers celebrate him as the "Hero of the beach!" (Several examples of the advertisment can be found on e.g. YouTube, For a general introduction see Wikipedia 2019). It appears unthinkable that present-day gyms and fitness chains would try to sell their products by telling how a punch in the face of your rival could restore a man's honor. Rather than promoting how the acquired strength can be used, it is the well-trained body as an aesthetic and sexual object that is advertised. That has appeal and sell tickets. This change, however, is perfectly understandable, since we are also biologically equipped to be attracted to beautiful bodies.

Studies across the world's regions and countries have shown that the features women now find attractive and beautiful in a man are the same ones which, in earlier societies, made him able firstly to protect a woman, and secondly to intimidate and defeat other men, e.g., (e.g., Etcoff 1999; Dixson et al. 2003; Maisey et al. 1999). Evidently, being tall and having broad shoulders, narrow hips, well-defined muscles, and a strong jawline is an advantage in love and war. Accordingly, men and women alike

find that a pear-shaped body with narrow shoulders and broad hips is the least desirable male figure. Although not all cultures place as much emphasis on muscle as we do in the West, no cultures see weak men as the most attractive (Gray and Ginsberg 2007; Maisey et al. 1999; Høgh-Olesen 2019). The backdrop is that for tens of thousands of years, humans lived as hunter-gatherers, depending on raw muscle power in our struggle to survive. During that era, cohabiting groups typically consisted of a couple of hundred individuals. Here, empathy, understanding, and the ability to protect one's group members were important. But it was equally important to be able to use one's strength and aggressiveness to defend the group and possibly to attack other groups (Diamond 1997)[6]. Male brawn was not just an advantage when hunting and killing prey to obtain the proteins needed to supplement the fruit-and-nut diet gathered by the group. Physical strength also enabled the group to protect its families against predatory animals and other men, which in turn enabled the women to concentrate on protecting the children. The difference in strength between men and women is larger in the upper body than in the lower body. Stones, clubs, and other weapons were thrown and wielded by hands and arms, making the men's greater upper-body strength an even greater advantage (Etcoff 1999, ch. 3). The musculature of the upper body is particularly sensitive to a surplus of testosterone in the blood, and usually, these very muscles (in the arms, shoulders, and chest) are the ones most vigorously trained by adolescent boys who join a gym. In other words, these young men are interested in increasing and reinforcing a gender difference that already exists. In this way, they hope to become more attractive to the girls. However, since the body at the same time sends a signal of brute force, the change also has the potential to transform their identity more fundamentally.

## 4. To Do Something and Become Someone

That some people involve themselves very actively in gym life can be understood as a way to create, develop, and work with their identity. At a superficial level, this commitment can be seen as the person's way of interpreting and expressing values, such as strength, health, dedication, youth, and sex appeal. These are values rooted not only in gym culture but also very broadly in the culture of the western world[7]. The values one identifies with and expresses through words and actions must, however, be accepted by one's surroundings before they can serve to construct an identity for the individual. Additionally, this acceptance should come from those we perceive as significant before it can be considered valid (Jenkins 2004). As an example, a person cannot just hang out in a bar with climbing shoes, harness, and carabineers and then expect that others will buy the postulate that they are a climber. The person may fool a few non-climbers in the bar, but the experienced climbers know that you are just posing. Climbers do not go to the bar with climbing gear. They use it on the mountain (Donnelly and Young 1999). Identity construction is, therefore, a dialectic process that consists of formulating and sending signals to the surrounding world, which interprets these signals and subsequently sends back a response. These things do not take place as separate processes but are an integrated part of daily life; a factor to which we rarely dedicate much reflection (Jenkins 2004). An activity is only really contributing to identity formation when the individual does something that others recognize.

The basic thought linking identity to recognition is the commonly held sociological idea that because of modernity, human beings are no longer known solely by virtue of our family, our property, or the place we grew up. Rather, we are who we are by virtue of our actions and our efforts—wherever these efforts may be applied (Giddens 1991). In reality, however, when it comes to education, employment, housing, and health, a person's socioeconomic background is still one of the best indicators of where they will end up in life (Cohen et al. 2010). Yet even despite this, the prevailing perception is that the

---

[6]　This is why it is erroneous to phrase the question of whether humankind is "fundamentally good or evil". Human beings evolved in an environment where, in order to survive, they had to have both aggressive and empathic traits.

[7]　The fact the fitness industry has been so successful—statistics show that about 60 million Europeans are members of a health club or gym (EuropeActive 2019)—is, among other things, due to its ability to market itself by these exact values.

recognition a person gets, which nurtures their identity, is a recognition that must be earned through their efforts in various contexts—and which we, therefore, cannot take over from our parents. As the German sociologist and philosopher Axel Honneth points out, we also receive recognition in the family, in the form of love and care, and legal recognition in the court system by virtue of our citizenship. However, these two types of recognition are not something a person can use to distinguish themselves in the social arena, genuinely setting them apart as someone special. This, however, can be done by virtue of a person's efforts, their skills, and their performance on the job and in their leisure time. The social recognition one can gain in this sphere is a very strong element in identity construction, precisely because it is conditional upon the efforts one expends and the results one achieves, and upon the recognition of these results by one's surroundings. That is what makes it qualitatively different from the unconditional recognition a person receives as love from the family, or recognition in the form of legal equality for a citizen of a nation-state (Honneth 1995).

All of this does not mean identity is a project isolated to the lone individual. Identity is oriented both inwards (towards the individual) and outwards (towards the world). Other people's definitions of who I am, through the way they treat me and what they say about me, is a crucial part of my self-image. Therefore, it is not enough for us to claim that we have an identity or to send out an identity signal. Such a signal must be accepted by the people in our surroundings, those we perceive as significant, before it can be considered valid (Jenkins 2004). It is, therefore, decisive to associate with others in order to become oneself.

## 5. Transforming the Body, Anchoring the Identity

People have different motives for training and becoming strong (e.g., Bates and McVeigh 2016; Cohen et al. 2007). Some scholars have argued that, in reality, bodybuilding is not for strong but weak men. Bodybuilding, says the American anthropologist Alan M. Klein, is compensatory behavior, and the muscular body should be understood as a "psychologically defensive construct that looks invulnerable but really only compensates for self-perceived weakness" (Klein 1993, p. 18). It is fair to apply a recently amount of healthy skepticism towards that kind of sweeping generalizations. Nonetheless, it is a fact that fitness training and lifting weights at the gym are not sports of the same sort as badminton, football, or swimming, where certain skills are developed, but rather an activity whose attraction lies in its promise of physical change and improvement. So, in addition to the defensive interpretation, where bodybuilding is viewed as an expression of psychological deficits, the activity can also pro-actively be understood in terms of the therapeutic effect the training and the transformation may have. This was how one of my interviewees, 22-year-old Martin, who at 161 cm is rather shorter than most other men, viewed it. He used his workouts to reinforce his self-image:

> "I'm not very tall, you know? So I thought: Well, if you're not tall, you've got to be wide instead. And there was this self-perception thing that came over me too. I didn't want to be lean and skinny. Plus, I've always been fascinated by people who were broad-shouldered and well-muscled."

Yet, Martin refuted the claim that the fascination with strength training could be reduced to a question of low self-esteem: "No, I've always had an easy time getting along with people and socializing, and I've never been the one who got pushed around or anything like that." Still, he acknowledged the therapeutic element, and he took no pains to hide his motives for working out:

> "It's a sport that changes your appearance. Not everybody needs to pump iron and get big to have self-esteem, but I feel like it bolsters my self-confidence to know that I do a sport that's visible, physically speaking. I needed something that could help my self-esteem. I see no reason to hide that. For me, that was important."

Martin used the concepts "self-esteem" and "self-confidence" interchangeably[8]. But the thrust of Martin's statements is that being 161 cm, he was certainly able to use weight-lifting to change his appearance and, consequently, to boost his self-confidence—and perhaps, in the longer term, his self-esteem. Contrary to Martin, Kasper (who was 178 cm and 21 years old when I spoke to him) had the experience of being bullied. He needed strength as a means to build self-confidence:

> "When I was sixteen, I weighed 65 kilos. Some guys were a lot bigger than me. Some guys still are. But I was also bullied a lot as a kid, so I think it's one of those inferiority complexes that first got me started."

Jesper, who was 27 years and 172 cm when I spoke with him, also talked about the issue of self-confidence as a pivotal motivator for him to begin lifting weights:

> "I was the little red-headed kid with glasses, and with braces and freckles and a pale complexion too, and I was easy prey for bullies. I wanted to change that. I got rid of my braces, and that was great, and then I wanted to do something about my size, weighing in at 58 kilos. So, I starting working out when I was eighteen."

Martin's, Kasper's, and Jesper's motives link in with the young man from the Charles Atlas advertisement. Additionally, it links with a theme regarding vengeance, which the Swedish doping scholar David Hoff has emphasized. In his interviews with users of AAS, he found that in relation to the issue of being bullied and harassed in school, several of his informants had been attracted to strength training and martial arts. It boosts self-confidence to know that you can defend yourself against physical assaults (Hoff 2016). Such reasoning corresponded with the motives of Kenneth, who was 24 years old and 180 cm when I spoke to him. The main object of his strength training was not his looks, but the desire to achieve a sense of security through strength. He needed this to deal with his insecurity in handling what he perceived as threatening social situations:

> "My main goal has always been self-defense, and that's probably because of this basic feeling of insecurity I've always had. And that's probably the main reason why I work out at all. Let's say I'm out somewhere, partying or whatever. Then if somebody starts acting like an asshole, I have an advantage in that; hopefully, I'll be a little bit bigger and stronger than that person. So my motivation is pretty much linked to the security in that."

In my informant group, Kenneth was the one who most clearly expressed personal insecurity as a motivation for his physical training. Although Kenneth was focused on his ability to defend himself, this need had not arisen out of personal experience in critical situations. When asked how many times he had been in a situation where he had to defend himself, he replied: "Not any, really. But it's more about the mental side of things. Because when I'm physically strong, I also feel mentally strong. It's probably just me, and the way it works on me." But Kenneth is hardly alone in such feelings. Being and feeling strong brings a feeling of calm and a sense of security that is also expressed in a person's body language. For one thing, the level of the stress hormone cortisol rises when a person feels inferior and holds a low position in the social hierarchy, whereas the level of testosterone rises when a person does well in competitions and is positioned at the top of the social hierarchy (Mazur 2005).

The strong body's appearance exhibits its potential for violence. This builds confidence in a person who is and feels strong, and at the same time, it is something people in the surroundings can recognize. Anton, who was 188 cm and 21 when we talked, can relate to that. He experienced how both reactions in the world around him and his self-confidence changed as he became larger and more muscular:

---

[8]   In the field of psychology, though, the two concepts are normally kept separate, with "self-esteem", denoting the basic feeling of value as a human being, which we have as part of our more or less stable core. "Self-confidence", on the other hand, is more superficial and can be affected by an experience of competence and praise from our surroundings in specific situations (Karpatschof and Katzenelson 2011).

"I've never been bullied, but it seems like I've always had low self-confidence when it came to girls and generally speaking. But then, I noticed, after I started working out, that my self-confidence just grew and grew. It's like you get stronger and stronger. And suddenly you're stronger than the people around you that you know, and suddenly you're bigger. And then, people start using a different tone of voice and respecting you. Things just changed for me. People's behavior changed. The way people talked to me changed. And then you get a whole different level of self-confidence."

The way Anton spoke about respect is in line with the description in this article's opening paragraph about strength and social hierarchies. We recognize that the response from people in the world around him is based on a basic biological impulse, grounded in fear of the consequences of violence. However, the more respect Anton gradually noticed as he grew was not a result of what he could actually do with his muscles, but rather on what he looked like he could do with his muscles. As noted, humans, like numerous other mammals, are able to read or decode social hierarchies without first testing them, for instance, in violent confrontation (Herbert 2015). This is part of the appeal of the muscular body: It plainly signals where a person is in that sort of primitive social hierarchy, which still greatly influences our perception of social situations, even though it no longer directly relates to our survival. Also, the self-confidence Anton described as accompanying his new position clearly affected him, and his perception of himself.

This also testifies to identity being variable rather than constant, and to our ongoing preoccupation with identification processes; of being something, and becoming someone, as argued by the British sociologists Richard Jenkins and the Canadian-American sociologist Erving Goffman (Jenkins 2004; Goffman 1959). Nevertheless, Anton's description indicates an aspect that Jenkins pays scant attention to, namely how important the body and body changes are to a person's identity. Jenkins sees the body as important, but in adults, he only deals thematically with changes accompanying the aging process. Goffman also refers to the body's relation to a person's surroundings and its importance to identity, but he focuses above all on the face (Goffman 1959). The significance of bodily changes for identity, as clearly evidenced in Anton's statement, is due not least to the way people in our surroundings react to the body we exhibit, thereby accepting and confirming the "new" identity, which can then take root.

Therefore, when Anton talks about the way those around him reacted to his bodily changes, it is not simply a misconception on his part. Experiments in social psychology have demonstrated how people tend to be more considerate to men with stronger upper bodies, treating them with greater respect. Two men with different physiques—say, one small and slight; the other big and strong—will, by comparison, develop different understandings of the world around them based on their different experiences during social interactions. Against this backdrop, they will develop different self-images and ways of interpreting future social situations (Kenrick et al. 2004). For instance, when men of slight build walk down a dark alley alone at night, they rarely find that people walking towards them have to cross the street to pass on the other side. Strong, tall, broad-shouldered men often see this happen. Such differences in experiences cause people to develop different self-images, different understandings of the world, and different learning strategies for how to act in different social contexts. Through the prism of their various experiences, people acquire different behaviors in various social situations. This is social learning grounded in concrete experiences, based on one's bodily appearance and physique, which is consistent with what Anton said about how his surroundings reacted to him as his physical-development project progressed.

Jesper had a similar experience with this. When asked whether only his perceptions had changed, or whether he also sensed changes in his surroundings as he grew, he replied without hesitation:

"Really, really big changes, from the people around me. When I first started being able to wear a tank top, and . . . I feel kind of like: when you look at yourself in the mirror, and you can see there's been a change, then you gain self-confidence, and that self-confidence is enough to make it just radiates out of you. And then when you've got enough of that radiation going, and when you've got an attitude . . . Just walking down the street with an

attitude—not exaggerating, but just having an attitude—well, that makes people look. And just having people look at you . . . "

This is a very accurate description of what a person who changes his body can experience in the dialectic process with their surroundings. In terms of identity, these experiences make a very strong impression because the person's image of who they are, or want to be, is confirmed by their surroundings. As we have seen, Martin and Anton's stories are very similar to Jesper's. This is how, through an ongoing dialectic process, larger muscles can change the type and the quality of a person's social interactions, and thereby also the person's body image, self-image, and identity.

Even so, the transformation to a more muscular body cannot only influence identity because of the potential for violence that it exhibits. In and for itself, violence has no evolutionary value. It is the result of the violence along with the ability to decode the potential for violence that has evolutionary value because these matters have consequences for a person's position in the social hierarchies and thereby his access to resources. Anton thus voiced how it was not only respect from his surroundings and a feeling of safety he attained, as he got larger. Girls also began to show him much more attention, which really was one of his original motives to begin lifting weights: "Yes, back then it was a real driver", he said and continued:

"I'd say, before, it would take a lot for me to be able to pick up a girl. That's certainly not a problem anymore. Quite the opposite, actually. Back then, it could take years for me to just start talking to a girl, but now, to be quite honest, it's almost like they won't leave me alone. But most girls just want me for my body."

This last comment was accompanied by a smile that clearly showed he knew the classic feminist accusation that men only see women as sex objects. Then, he elaborated:

"You know, it's so plain to see what they're interested in. I know it wouldn't have been like that if I hadn't worked out. Everybody says, "it's not your looks that count", but you're lying if you say looks aren't important."

Clearly, Anton has experienced how girls gave him increased attention after he became more muscular. He has become more attractive as a man. Naturally, that also affects his self-image, self-confidence, and identity. Nevertheless, as regards shaping an identity as someone who is strong and has skills in lifting weights, the attention from girls is perhaps less important than the recognition the person receives from others who are also serious about the activity. That, at least, is how Kasper assessed it. He agreed to emphasize sexual charisma as important, but that does not mean that women's compliments are more important than men's:

"I prefer to get compliments from a guy who works out himself, and who knows what he's talking about. Let's say I'm at a club, and some girl walks over and lays her hand on my chest. That doesn't mean anything, because she hasn't thought about how much work it represents. Sure, it's nice and all. But that's not what makes me tick. But if somebody walks over, maybe even a pro, who says things are looking good. I mean, if the alpha male gives you a pat on the back, you're doin' pretty good."

Kasper's final comment illustrates that identity is not something that can be performed, but something that requires actions that are recognized by the right people. That was the point in the above example with the wannabe-climber. Recognition has a much more profound effect when it comes from those who have knowledge about the issue. The fact that girls think you are hot is comforting and can lead to a great night out, but, as in all other contexts, the most important recognition comes from those with status in the culture.

From research, we know that recognition and social status influence overall health in a positive way (Cohen 2004). In that sense, there might be a health benefit associated with the recognition one can attain in the gym. However, perhaps there is also a different kind of health at play when striving to win a muscular body and by that move upwards in the social hierarchy.

## 6. A Different Kind of Health

Anton, Martin, and Jesper had all used AAS to become bigger and stronger. Kenneth and Kasper had not, but they trained with peers that used them and were frank about the temptation to use. One may hold the view that social relations, our perceptions of attractive bodies, and the sexually charged gendered interaction alone are based on social conventions. In that case, however, the use of AAS is a bigger mystery than one may be willing to admit. Explanations based primarily on social constructivist theory have been tested, but they either lack Popper's criterion of falsifiability or require far more assumptions and supporting premises than the evolutionary psychological perspective presented above. If one applies the principle of Ockham's razor, that is the idea that one should select the explanation that requires the fewest assumptions when trying to understand something, then the evolutionary psychological perspective is far more 'thought-economical' if one wants to understand the use of AAS. Obviously, culture plays its part, and the market and the media affect our preferences significantly. Meanwhile, the market and the media would not be able to force any values on culture if the people in that culture would be unreceptive to those values. Being strong and muscular was once a biological, evolutionary advantage. That biological advantage became an aesthetic preference (Etcoff 1999). Moreover, the evolutionary perspective also has explanatory power if we want to understand the health risks young men run when using AAS.

Besides a number of acute side effects, the use of AAS also increases the risk of heart failure and, thereby, premature death (Pope et al. 2014; Rasmussen et al. 2018). Depending on how intense a regime the user has applied in his 20s and 30s, he risks dying in his late 50s rather than in his late 70s. To lose 20 years of one's life appears to be a severe deterrent, so why would young men still choose to use AAS?

We have already seen how the difference between having a weak and feeble body versus a strong and muscular one may be associated with being positioned in the bottom or at the top of the social hierarchy. From an evolutionary perspective, an individual's position on the social hierarchy is tantamount to access to resources, which is not just food and protection but also power, recognition, and possibilities for mating. The duration of time, it takes a human being to grow up, mature, reproduce, and make sure that one's offspring can grow up and repeat the whole thing is approximately 45–50 years. For a fruit fly, it is about eight days; for rats and mice, it is one year; and for cows and horses, 15 years. Biogerontologists, like the Danish-Indian researcher Suresh Rattan, have termed such duration of time the "essential lifespan" of a species as required by evolution. To live longer than the "essential lifespan" of one's species is a privilege, not a right, Rattan explains, because nature never intended us to be much older than our essential lifespan (Rattan 2018).

The heretical thought this perspective gives rise to is that when young men, like Anton above, runs a health risk by using AAS to obtain a more attractive body, they do so for the sake of a different kind of health. Anton and his peers may be acting irrationally if judged with a blind premise that one should seek to get as old as possible, but rational seen from a perspective where an individual has the biological impetus to try and obtain the best possible position that will make the person thrive inside his essential lifespan. As Anton said about his experience with his position in the social hierarchy when he got larger: "And then people start using a different tone of voice, and respecting you. Things just changed for me. People's behavior changed. The way people talked to me changed". Likewise, on his relationship with girls before and after he took AAS: "Back then, years could pass before I got in contact with a girl, but now I can honestly say they won't leave me alone".

In the larger perspective, behavior that leads to such scenarios are rational—they promote a different kind of health. Anton and his peers do not need to know anything about evolutionary psychology or the essential lifespan of a species to experience the advantages a muscular body gives. The principles of evolution manifest themselves in our psyche and behavior regardless of our knowledge of them. Yet, when we know these principles, we are in a better position to understand what on the surface may appear irrational.

## 7. Closing Remarks

Human beings are not like other animals. While issues of dominance and social hierarchies are exposed under rather similar conditions in nature, among humans, they are largely displayed in culture. We may be exposed to some of the same stimuli, but we have many more layers of sophisticated behavior in which to respond. Take testosterone as an example: The hormone is responsible for the development of the physical features that typify the male body, like muscles, growth of beard, and a more pronounced jawline, and it plays a significant role in male's greater inclination to take part in dominance-based competitions (Herbert 2015). What is more, even though humans and animals are under the influence of testosterone, human beings have become ever more peaceful as civilization has progressed, whereas the use of violence in the animal kingdom is the same today as it was millennia ago (Pinker 2011). A consequence of the civilizing process is that status and resources are no longer won with a clenched fist, but in social communities, in the leisure time, and on the job market. Correspondingly, the most recognized types of status are not displayed by muscles and brute force, but rather in symbols like education, job position, and consumer goods. Civilization has thus changed our behavior, but evolution still influences our psyche and preferences.

This is the reason why appearance, muscles, and strength continue to play a significant role. Not least for young men for whom cultural dominant status symbols are not directly accessible. They build their bodies because the concrete physical transformation also changes their relationship and interaction with the world around them. This happens partly because of the indication for the potential for violence it exhibits, and which others can read directly from the strong body, and partly because of the beauty a well-trained body radiates. These issues build identity in a way few scholars have recognized. At the same time, they are crucial in order to understand why working out in the gym is so attractive, and it assists our understanding of why some young men are tempted to supplement their workout with AAS to obtain results faster.

**Funding:** There was no funding associated to this project.

**Conflicts of Interest:** No conflicts of interest to declare.

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
