# Peer review of "Civilized Muscles: Building a Powerful Body as a Vehicle for Social Status and Identity Formation"

_socsci, doi:10.3390/socsci8100287_

Round 1

Reviewer 1 Report

Would like to see more set-up/introduction to the opening lines, "He who can beat..."

p.1 line 26- "It"-- what/who does "IT" refer to? This entire paragraph could be revamped to be clearer.

p.2- line 46 Theory and Method: need additional references in first paragraph. Need to be more specific,line 51: "This" ? and line 53, "This is the case..." ?

p.2- participant breakdown requires more information:  Are they Caucasian? Black? Heterosexual? Training for a specific sport?

p.3- line 91 (and paragraph that follows) - requires referencing/citations

p.3 - line 111 and on requires referencing/citations

p. 4- line 125- could you cite the actual advertisement/Youtube link? vs. Wikipedia.

p. 4- line 130 "Studies across the world's ..." what studies, include references/citations

p.8 - discussion form participants on receiving extra attention from women. Assume all participants were heterosexual? There is fairly extensive literature on the culture of muscle building/gym going/bodybuilding and homoeroticism that may be relevant to your analysis. For example, see:

Halkitis, P. N., Moeller, R. W., & DeRaleau, L. B. (2008). Steroid use in gay, bisexual, and nonidentified men-who-have-sex-with-men: Relations to masculinity, physical, and mental health. Psychology of Men & Masculinity, 9(2), 106-115.http://dx.doi.org/10.1037/1524-9220.9.2.106   Nicholas Lanzieri LMSW & Tom Hildebrandt PsyD (2011) Using Hegemonic Masculinity to Explain Gay Male Attraction to Muscular and Athletic Men, Journal of Homosexuality, 58:2, 275-293, DOI: 10.1080/00918369.2011.540184    Additional sources you should consider including:   "Making Muscle Junkies": Investigating Traditional Masculine Ideology, Body Image Discrepancy, and the Pursuit of Muscularity in Adolescent Male Source: International Journal of Men's Health . Fall2011, Vol. 10 Issue 3, p220-239. 20p. MARTIN, JARRED; GOVENDER, KAYMARLIN    Signe Ravn & Julia Coffey (2016) ‘Steroids, it’s so much an identity thing!’ perceptions of steroid use, risk and masculine body image, Journal of Youth Studies, 19:1, 87-102, DOI: 10.1080/13676261.2015.1052051     Nicole Thualagant (2012) The conceptualization of fitness doping and its limitations, Sport in Society, 15:3, 409-419, DOI: 10.1080/17430437.2012.653209

Author Response

I would like to thank the two reviewers for their positive and constructive comments to my work. I believe that with the amendments I have made as a response to the comments, the article has improved. I hope the editors and reviewers agree. The most significant changes are in the paragraph now titled ‘Method and theoretical background’, which has been expanded so it now has a general introduction setting up a framework for the article and then there is a substantial addition on methods and interviews.

Below, each of the reviewers’ comments is presented in the left side column and my response to the comment in the right side. Line numbers refer to the revised manuscript. As the second reviewer had some more general comments, I will begin with those.

Reviewers comment

Author’s response

Comments numbered

Reviewer 2

1

This shows potential. You have some great ideas and I like the premise but I don't think you have convinced me as well as you could

Thanks you. I have restructured the introduction to make it clearer where the paper, the results and thinking comes from. Line 55-67

2

Methods need expanding - why only 5 participants? What was your research question? How did you recruit participants?

Now expanded. Line 68-85.

3

Findings seem a bit thin and out of proportion with the literature. It seems that most of the paper is background and discussion and that the findings lack a bit of substance – can you go deeper into your findings?

I believe I am familiar with the literature in this field. The aim of the article is to look into an area that has not had much attention. The aim, purpose and intention of the paper is now made clearer in line 55-67.  The article is not to be considered an interview study and therefore the aim is not to go deep into the empirical findings (I do that elsewhere (Christiansen, forthcoming (2020))), but rather to investigate one element in the appeal of the muscular body, that hitherto has not been given much attention in the academic literature.

4

While an evolutionary perspective is interesting I am not sure that you have demonstrated the best fit with your findings.

I am not quite sure what to do with this comment. Not examples are provided to illustrate where the findings do not fit with what I say about evolutionary psychology. Perhaps the amended introduction makes my use of the interview quotes clearer.

5

I am not sure that you have done a comprehensive review of the literature in this area before writing - what about the other papers that look at muscle from an evolutionary perspective? Does the 'crisis in masculinity' theory explain what you are finding?

I do know the literature quite well. I am aware about a number of studies that do look into this, but they have a different take. Should I introduce the article with a comprehensive literature review, it would either be a very long or completely different article.

6

You have used secondary sources and Wikipedia - this is not acceptable at this level. Go to the original sources.

This is an odd comment. I get the suspicion that the reviewer only saw that there was a wiki reference but did not check what it was a reference to. It is a reference to an advertisement that ran in comic books and magazines in the 1940s, 40s, 50s and into the 60s. It is an advertisement, so there is no original source to go to. The best source I could find is the Wikipedia reference. There are a number of podcasts and small documentaries available on YouTube. But it would not be fair to choose any one between them. So, while I have added a short explanation, I will stick to the Wikipedia reference.

7

Expression is a bit awkward and unclear in numerous places and there are numerous grammatical error (I have noted many of these below) and I suggest editing by someone with a better grasp of English so that these things don't cloud your ideas (which are essentially good).

A native English speaker has read and checked the paper.

8

Is page 1 lines 22-25 a quote?

No. An opening (thought-provoking) statement meant to set the scene for the article’s topic.

9

Page 1 line 37 has been (not have been)

Thanks. Amended.

10

Page 2 line 53 expression is awkward here (and in other places throughout the paper): or coming to look strong

Thanks. Amended. (“and why there is still an attraction in becoming strong and muscular”)

11

Line 65 – incomplete sentence: conducted 14 long, semi-structured interview with.

Thanks. Fixed

12

Lines 74-5 awkward expression: “on the one hand side and our ability to exert self-control and delay of gratification on the other”.

Amended

13

Page 2 note 4 should read “mutually dependent”

Thanks. Amended.

14

Line 108: ‘was’ not ‘were’

Thanks. Amended.

15

Line 125: Wikipedia! This is not an acceptable academic source

See comment 6

16

Line 130 – which studies?

Those mentioned in line 137 in the original and 179 and 184 in the revised

17

Line 155 typo in something

Thanks. Fixed.

18

Line 166 awkward expression

Fixed

19

Line 181 an extra ‘the’ before sociologist

Deleted

20

Line 184 are not

Fixed

21

Line 365 have knowledge

Fixed

22

Line 401 have not has

Fixed

23

Line 428 missing references at the end of the sentence?

Fixed – reference (Herbert 2015) moved.

Reviewer 1

24

Would like to see more set-up/introduction to the opening lines, "He who can beat..."

Hmm, I rather like it this way. It sets the scene for the article and attracts the reader’s attention. I have added an introduction in the next paragraph (line 55-67).

25

p.1 line 26- "It"-- what/who does "IT" refer to? This entire paragraph could be revamped to be clearer

“It” refers to the use of violence and the above description. However, I have rephrased the sentence to make it clearer.

26

p.2- line 46 Theory and Method: need additional references in first paragraph. Need to be more specific,line 51: "This" ? and line 53, "This is the case..."

More references are added.

The sentence reads: “evolution has equipped us with a psyche that is receptive to the signal the strong body transmits to its surroundings. This is why we treat stronger men with greater respect than weaker men”. I find it clear that “This” refers to the previous sentence and the psyche evolutions has equipped us with.

Similar with “This is the case” – which refers to the attraction of becoming strong. No amendments are made.

27

p.2- participant breakdown requires more information:  Are they Caucasian? Black? Heterosexual? Training for a specific sport

More info added line 68-85.

28

p.3- line 91 (and paragraph that follows) - requires referencing/citation

References added.

29

p.3 - line 111 and on requires referencing/citation

From line 109 (original manus), it is explicated that the following is from Stephen Pinker’s book The Better Angels of Our Nature – A History of Violence and Humanity. I have added the reference earlier on, line 154 in revised manus.

30

p. 4- line 125- could you cite the actual advertisement/Youtube link? vs. Wikipedia

See comment 6. Wikipedia is the best and most extensive reference here. I have added an explanation.

31

p. 4- line 130 "Studies across the world's ..." what studies, include references/citation

The studies are referenced in line 137 in the original (179 in the revised). I have added a few more references.

32

p.8 - discussion form participants on receiving extra attention from women. Assume all participants were heterosexual? There is fairly extensive literature on the culture of muscle building/gym going/bodybuilding and homoeroticism that may be relevant to your analysis.

This is an interesting suggestion. I am aware of this body of literature, but my assessment is that, introducing this would be outside the scope of the present article. I have now emphasised in the methods section that all interviewees identified as heterosexual.

33

34

35

36

37

Reviewer 2 Report

This shows potential. You have some great ideas and I like the premise but I don't think you have convinced me as well as you could.

Methods need expanding - why only 5 participants? What was your research question? How did you recruit participants?

Findings seem a bit thin and out of proportion with the literature. It seems that most of the paper is background and discussion and that the findings lack a bit of substance - can you go deeper into your findings?

While an evolutionary perspective is interesting I am not sure that you have demonstrated the best fit with your findings.

I am not sure that you have done a comprehensive review of the literature in this area before writing - what about the other papers that look at muscle from an evolutionary perspective? Does the 'crisis in masculinity' theory explain what you are finding?

You have used secondary sources and Wikipedia - this is not acceptable at this level. Go to the original sources. 

Expression is a bit awkward and unclear in numerous places and there are numerous grammatical error (I have noted many of these below) and I suggest editing by someone with a better grasp of English so that these things don't cloud your ideas (which are essentially good).

Other minor comments: 

Is page 1 lines 22-25 a quote?

Page 1 line 37 has been (not have been)

Page 2 line 53 expression is awkward here (and in other places throughout the paper): or coming to look strong

Line 65 – incomplete sentence: conducted 14 long, semi-structured interview with.

Lines 74-5 awkward expression: “on the one hand side and our ability to exert self-control and delay of

gratification on the other”.

Page 2 note 4 should read “mutually dependent”

Line 108: ‘was’ not ‘were’

Line 125: Wikipedia! This is not an acceptable academic source

Line 130 – which studies?

Line 155 typo in something

Line 166 awkward expression

Line 181 an extra ‘the’ before sociologist

Line 184 are not

Line 365 have knowledge

Line 401 have not has

Line 428 missing references at the end of the sentence?

Author Response

(The authors gave the same response as above.)

Round 2

Reviewer 2 Report

I appreciate the efforts the author/s has/have gone to to address my concerns. These changes have greatly strengthened the paper, particularly by making clear the aims and the place of this paper in the larger project, and by improving some instances of awkward expression.

I still think that the previous literature that takes an evolutionary perspective on muscle should at least be acknowledged, and that the author/s should explain how their paper differs/adds to this previous work, but I leave this to the editors to decide.

I am guilty of not checking the reference to Wikipedia in my haste and appreciate the clarification. 

Well done, i think this makes an important addition to understanding.